# Trading Carbon for Physics: On the Resource Efficiency of Machine Learning for Spatio-Temporal Forecasting

## Abstract

Development of modern deep learning methods has been driven primarily by the push for improving model efficacy (accuracy metrics). This sole focus on efficacy has steered development of large-scale models that require massive resources, and results in considerable carbon footprint across the model life-cycle. In this work, we explore how physics inductive biases can offer useful trade-offs between model efficacy and model efficiency (compute, energy, and carbon). We study a variety of models for spatio-temporal forecasting, a task governed by physical laws and well-suited for exploring different levels of physics inductive bias. We show that embedding physics inductive biases into the model design can yield substantial efficiency gains while retaining or even improving efficacy for the tasks under consideration. In addition to using standard physics-informed spatio-temporal models, we demonstrate the usefulness of more recent models like flow matching as a general purpose method for spatio-temporal forecasting. Our experiments show that incorporating physics inductive biases offer a principled way to improve the efficiency and reduce the carbon footprint of machine learning models. We argue that model efficiency, along with model efficacy, should become a core consideration driving machine learning model development and deployment.[1]

## 1 Introduction

Machine learning (ML) has advanced rapidly over the past decade (Sevilla et al., 2022), driven largely by the pursuit of improving model efficacy (accuracy metrics). This narrow focus has led to increasingly large models that require substantial computational resources and generate significant carbon emissions across their lifecycle (Strubell et al., 2019; Anthony et al., 2020; Luccioni et al., 2023). As models scale, efficacy gains diminish while costs escalate, underscoring the need to move beyond accuracy as the sole measure. A more holistic assessment must balance efficacy with efficiency, capturing not only predictive performance but also compute, energy, and carbon costs (Henderson et al., 2022). This requires exploring trade-offs and identifying strategies that strike a favourable balance, rather than lopsided optimization.

Physics-informed ML offers a promising way to achieve this balance. Physics priors such as conservations laws, symmetries, or governing equations can be embedded into model design through, e.g., loss terms, architectural layers, or mechanisms. This combines the flexibility of data-driven methods with the structure of domain knowledge (Fig.1-a). As shown in Fig. 1-b, a semi-supervised physics-informed neural network (PINN) achieves the best accuracy–carbon trade-off, outperforming both a purely data-driven neural network (NN) and a physics-only PINN (see Sec. 4.1 for details). This example highlights how hybrid approaches can yield superior efficiency without sacrificing efficacy.

Building on this intuition, we extend the perspective from task-specific models such as PINNs to general-purpose architectures that embed physics inductive biases in ways that are transferable across datasets and domains. We focus on spatio-temporal forecasting of partial differential equations (PDE) dynamics, a setting characterised by high computational cost and practical relevance. Operational weather and climate forecasting, for instance, requires large-scale PDE solvers running

---

[1]Source code for the experiments in this work are available at: `https://anonymous.4open.science/r/FlowMatching-DC16/`.

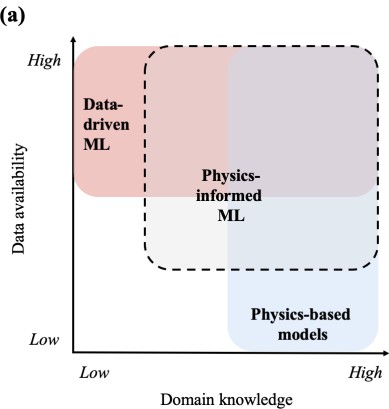

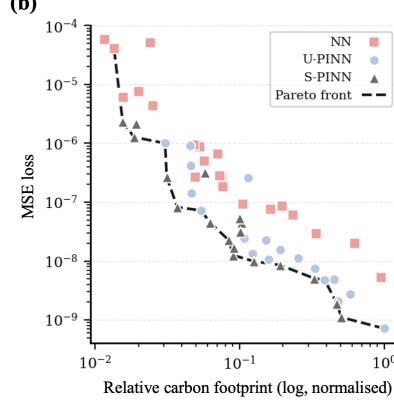

Figure 1: **a:** Conceptual map of modelling approaches by data availability (vertical axis) and domain knowledge (horizontal axis). Physics-informed ML lies in the middle, combining data-driven flexibility with physics priors. **b:** Test MSE versus $CO_2$eq emissions for a data-driven NN (red squares), an unsupervised PINN (blue dots), and a semi-supervised PINN (grey triangles). The dashed line marks the Pareto front, dominated by the S-PINN, indicating the best accuracy-carbon trade-off. Details of the experiment are in Sec. 4.2.

on supercomputers multiple times per day (Lam et al., 2023). In such contexts, even modest efficiency improvements can translate into substantial operational carbon savings, making this domain an ideal test-bed for exploring sustainable ML design (Van Wynsberghe, 2021). Our key contributions towards this end are:

1. **Carbon-aware evaluation:** We promote evaluation practices that consider both predictive accuracy and carbon cost, explicitly characterising trade-offs.

2. **Characterising the accuracy–carbon trade-off:** We conduct three experiments on PDE dynamics and spatio-temporal forecasting to compare a range of models with varying levels of physics inductive bias. Our main study focuses on forecasting incompressible shear flow, comparing U-nets (no/weak bias), flow matching (medium bias), and Fourier neural operators (strong bias).

## 2 BACKGROUND AND RELATED WORK

**Revisiting the Bias-Variance Trade-off.** It is commonly understood in ML that strong model assumptions (bias) risk under-fitting, whereas high sensitivity to training data can lead to overfitting (variance) (Kohavi et al., 1996). Striking the right balance is crucial to obtain the right class of models.

The same trade-off can also offer a useful way to perform model selection when viewed through the lens of resource consumption. Typically, if domain knowledge (physics-based or others) can be incorporated into models as inductive bias, it can reduce the reliance on training data or reduce prediction errors. This is illustrated in Fig. 2 for the task of modelling a harmonic oscillator. Models with appropriate inductive bias (sinusoidal oscillations) perform better for the same number of training data points and model parameters. See Sec. 4.1 for more details.

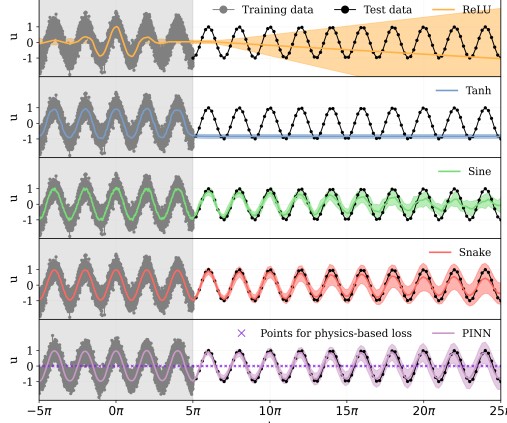

Figure 2: Modelling the harmonic oscillator. Training and extrapolation predictions when using NNs with different activation functions: `ReLU`, `Tanh`, `Sine`, `Snake`, and using a PINN.

**The Question of Resource-Awareness.** The standard metrics used to characterise the performance of ML models primarily focus on efficacy and are untethered from efficiency considerations. This introduces a preference for models that could offer high efficacy at large resource costs (Bakhtiarifard et al., 2024). There are two primary streams of work that aim to include resource-aware metrics: using composite metrics and multi-objective optimisation.

Several composite metrics have been proposed to combine predictive performance with resources like energy consumption or carbon footprint. Evchenko et al. (2021) introduced a composite metric that combines efficacy and efficiency metrics to study the influence of resource constraints on traditional (non deep learning) ML models. *Carburacy* (Moro et al., 2023) extends the idea to transformer models by jointly quantifying accuracy and carbon emissions. More recently, Kapoor et al. (2025) proposed *EcoL2*, a composite metric similar to *Carburacy* but extended to include the costs of data generation and hyperparameter tuning.

Multi-objective optimisation has been primarily used in neural architecture search (NAS) resulting in models like EfficientNet (Tan & Le, 2019). More recently, energy consumption (Bakhtiarifard et al., 2024) and carbon footprint (Zhao et al., 2024) have been taken into consideration in NAS. These methods, however, have not actively studied the influence of physics inductive biases on the resource consumption of ML models.

**Spatio-Temporal Forecasting Models.** There are many model classes for the task of spatio-temporal forecasting; each differing in the extent of physics inductive bias they use. Fig. 3 provides a high-level overview of this span for models considered in this work.

*U-net.* U-nets (Ronneberger et al., 2015) can be used for spatio-temporal forecasting by mapping a window of past states to future ones. They capture local correlations by fusing features across multiple spatial scales, but do not explicitly constrain the dynamics beyond what is enforced by the training data. Mild forms of

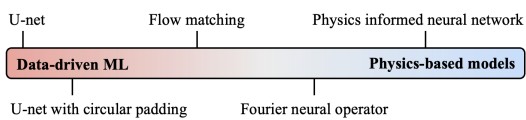

Figure 3: Spectrum of model families from purely data-driven ML to physics-based solvers.

inductive biases can be introduced to convolutional architectures like U-net by including circular padding to respect periodic boundaries and avoid edge discontinuities.

*Flow Matching.* Recent classes of probabilistic generative models like flow matching (FM) (Lipman et al., 2023) can be viewed as an intermediate class of models that can include some degree of inductive bias by modelling dynamics as continuous-time flows. This formulation resonates with dynamical systems and intuitively suggests smoother trajectories. FM has been applied to spatio-temporal forecasting and other PDE dynamics tasks (Lim et al., 2025; Li et al., 2025), with Baldan et al. (2025) further embedding PDE residuals into the objective.

*Fourier Neural Operator.* The Fourier Neural Operator (FNO) (Kovachki et al., 2023) introduces a physics prior through its Fourier-space representation, enabling the model to capture global spectral interactions. This makes it particularly well-suited to systems with periodicity and long-range correlations. In our spatio-temporal forecasting setting, we adapt the FNO from its original operator-learning formulation (Lu et al., 2021) to an autoregressive setup, where its spectral representation provides a strong inductive bias compared to the U-net by explicitly encoding global and periodic structure.

*Physics-Informed Neural Network.* PINNs (Raissi et al., 2019) embed governing PDEs directly into the training objective, thereby regularising the optimisation landscape and guiding solutions towards physical consistency. By minimizing PDE residuals, PINNs can even be trained without supervised data, representing the strongest inductive bias among the models considered. Embedding conservation laws, symmetries, or governing PDEs into the ML model development pipeline has been shown to reduce the number of trainable parameters (Dutta et al.; Patra et al., 2024), reduce the amount of training data (Psaros et al., 2023; Zhong et al., 2021), and accelerate convergence (Brehmer et al., 2024; Jahani-nasab & Bijarchi, 2024).

## 3 METHODS

**Problem Setup.** We consider spatio-temporal forecasting of dynamical systems in the general setting, where the objective is to approximate the solution $u(x, y, t)$ of a PDE given basic inputs

such as initial and boundary conditions. We frame this as an autoregressive prediction task on a two-dimensional, uniformly spaced mesh $(x, y) \in \{x_1, x_2, \ldots, x_H\} \times \{y_1, y_2, \ldots, y_W\}$ with discretized time steps $t \in \{t_1, t_2, \ldots, t_T\}$. At each step $t_k$ the state is represented as a tensor $u_k(x, y) \in \mathbb{R}^{H \times W \times C}$, where $k \in \{1, 2, .., T\}$ indexes discrete time steps in a trajectory of length $T$. $H$ and $W$ are spatial dimensions and $C$ is the number of physical fields (channels), such as pressure or velocity components.

To capture temporal dependencies, the model is conditioned on a finite history of length $h$. The input window is denoted as $u_{k-h+1:k}(x, y) := (u_{k-h+1}(x, y), \ldots, u_k(x.y))$. A parameterised function $f_\theta$ is trained to predict:

$$u_{k+1}(x, y) = f_\theta(u_{k-h+1:k}(x, y)). \tag{1}$$

During training, the loss is computed on one-step predictions. At inference, we assess long-term stability via autoregressive rollouts, obtained by iteratively applying Eq. 1 and feeding the model's own predictions back as inputs.

We adapt this generalised model formulation to three problems. Each problem is then tackled with relevant set of methods that range from being entirely data-driven to offering relevant degrees of physics inductive biases.

1. **Harmonic Oscillator.** Modelling the dynamics of the harmonic oscillator only requires the time variable i.e, $u(t)$. For the harmonic oscillator, Eq. 1 reduces to $u_{k+1} = f_\theta(u_{k-h+1:k})$, where the spatial variables $(x, y)$ are dropped.

   We assess the impact of appropriate physics-inductive biases by comparing simple feed-forward NNs with and without biases. Specifically, these biases are introduced through activation functions and a loss term, for modelling the damped and undamped oscillators.

2. **Viscous Burgers' Equation.** Solving the viscous Burgers' equation increases the modelling complexity compared to the harmonic oscillator, as it has one spatial and one temporal variable to be considered. Eq. 1 for this case is given as: $u_{k+1}(x) = f_\theta(u_{k-h+1:k}(x))$, where only one spatial variable $x$ is modelled along time.

   We evaluate how different combinations of data and physics constraints influence predictive accuracy and efficiency by comparing a supervised NN, a physics-only PINN, and a hybrid PINN for approximating the viscous Burgers' equation.

3. **Incompressible Shear Flow.** Modelling the spatio-temporal dynamics of incompressible shear flow governed by Navier-Stokes equations is the most complex task considered. It has dependence on both the spatial and temporal variables. As a result, we use Eq. 1 in its original formulation.

   We evaluate several models spanning a broad spectrum of physics biases, depicted in Fig. 3. Purely data-driven models like U-net, the moderately biased model FM, and strongly biased neural operators are adapted for this task. The U-nets and FNOs use a history of $h = 4$ states, i.e. they are trained for one-step prediction according to Eq. 1 i.e., $u_{k+1}(x, y) = f_\theta(u_{k-3:k}(x, y))$.

   FM replaces direct one-step prediction with a continuous transport between consecutive states. Instead of learning a map $u_{k-h+1:k}(x, y) \mapsto u_{k+1}(x, y)$, FM predicts the next state by transporting the current state forward along a learned flow. Starting from $u_k(x, y)$, the method defines a flow trajectory $\phi_\tau(u_k)$ over a synthetic time axis $\tau \in [0, 1]$, where the flow evolves under a time-dependent vector field $v_\tau$. The one-step forecast is then obtained as the endpoint of this flow: $\hat{u}_{k+1}(x, y) = \phi_1(u_k(x, y))$, where $\partial_\tau \phi_\tau(u) = v_\tau(\phi_\tau(u)), \phi_0(u) = u_k(x, y)$. In practice, a NN is trained to approximate this vector field, defined by an interpolation path between consecutive states. Further FM details including the interpolation path, vector field, and learning objective are provided in App. A.

## 4 Data and Experiments

Based on the problem formulation in Sec. 3, we set-up experiments on the three tasks to explore the trade-off between efficacy and efficiency. Specific choices for each of the experiments are elaborated in detail next.

## 4.1 HARMONIC OSCILLATOR

We use simple feed-forward NNs to model the harmonic oscillator by introducing physics biases in two ways: through activation functions, where `Sine` and `Snake` (Ziyin et al., 2020) have periodic biases, and through a physics-informed loss objective. `ReLU` and `Tanh` serve as unbiased baselines.

The training data is generated from the damped harmonic oscillator, $u(t) = e^{-\delta t}(2A\cos(\phi + \omega t))$, where: $A = 1/(2\cos\phi)$, $\phi = \arctan(-\delta/\omega)$ and $\omega = \sqrt{\omega_0^2 - \delta^2}$, with added Gaussian noise ($\mu = 0, \sigma = 0.3$). We consider both the non-damped ($\delta = 0$) and the damped ($\delta = 0.02$) regimes.

All models use the same underlying architecture (two hidden layers of width 64), trained for 100 epochs with Adam optimizer (Kingma & Ba, 2015) using a batch size of 25. Learning rates are 0.01 for standard networks and 0.001 for the PINN, which employs `Snake` activations. Results are averaged over 10 runs. Additional details are provided in App. B.

## 4.2 VISCOUS BURGERS' EQUATION

In this experiment, we study models to approximate the solution of the viscous Burgers' equation with Dirichlet boundary conditions on a 2D spatio-temporal grid:

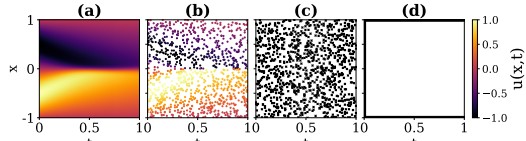

$$\frac{\partial u}{\partial t} + u\frac{\partial u}{\partial x} - \nu\frac{\partial^2 u}{\partial x^2} = 0,$$
$$u(x, 0) = -\sin(\pi x),$$
$$u(1, t) = u(-1, t) = 0.$$

Figure 4: Target and training data for viscous Burgers' equation. **a:** Target data. **b:** Supervised samples. **c:** PDE residuals samples. **d:** Initial and boundary condition points.

for $x \in [-1, 1]$, $t \in [0, 1]$, and with fixed viscosity ($\nu = 0.025$). The domain is discretized with 256 spatial points and 100 time steps. We compare three models with varying physics inductive biases, all using the same underlying architecture (comprising four hidden layers, width 50, `tanh` activation functions):

- NN: a supervised NN trained with the mean-squared error (MSE) on ground-truth data, using the loss objective: $\mathcal{L}_{\text{NN}} = \mathcal{L}_{\text{data}}$.
- U-PINN: an unsupervised PINN trained only on physics constraints that minimizes PDE residuals, initial conditions (IC) and boundary conditions (BC). Its loss function, $\mathcal{L}_{\text{U-PINN}}$, is composed of the residuals of the PDE and the deviations from the initial/boundary conditions, expressed as: $\mathcal{L}_{\text{U-PINN}} = \mathcal{L}_{\text{PDE}} + \mathcal{L}_{\text{IC}} + \mathcal{L}_{\text{BC}}$.
- S-PINN: a semi-supervised PINN where half of the training points are used for PDE residuals and boundary/initial conditions, while the other half enforce a data-driven penalty (MSE). Its loss objective is given as, $\mathcal{L}_{\text{S-PINN}} = \mathcal{L}_{\text{U-PINN}} + \mathcal{L}_{\text{NN}}$.

The specific formulations of all three loss objectives are provided in App. C. All models were trained using Adam optimizer followed by L-BFGS optimizer (Liu & Nocedal, 1989) with learning rate $10^{-3}$. Dataset sizes range from $10^3$ to $5 \times 10^3$ training points for PINNs and up to $8 \times 10^3$ for the NN. Data and code are adapted from Khoa (2022).

## 4.3 INCOMPRESSIBLE SHEAR FLOW

We set-up the large-scale experiment on modelling incompressible shear flow governed by the Navier–Stokes equations, using the spatio-temporal forecasting models introduced in Section 2.

**Data.** The incompreesible shear flow data is derived from *The Well* dataset (Ohana et al., 2024) generated with *Dedalus* (Burns et al., 2020). We use a subset of the full dataset consisting of 240 simulations spanning six PDE parameter settings with 40 initial conditions each. Simulations are resolved at $256 \times 512$ with 200 time steps ($\Delta t = 0.1$). The data is split by initial condition into 80/10/10 for training, validation, and testing, yielding $37,632$ one-step samples for training, $4,704$ for one-step inference, and 24 trajectories for autoregressive rollout (initialised at $t = 0$). The dataset is particularly challenging due to the coupled evolution of four physical fields (velocity components,

pressure, tracer) and its non-linear spatiotemporal dynamics at high resolution. Further details on PDE formulation and initialization are provided in App. D.

**Models.** We evaluate eight architectures spanning a spectrum of physics inductive biases, from purely data-driven to strongly physics-informed, summarised in Table 1. We use standard U-nets (Ronneberger et al., 2015) and a ConvNeXt variant (UNet-CN) (Liu et al., 2022) for improved hardware-efficiency. Zero-padding serves as our no-bias baseline, while circular padding (CP) introduces a weak

| Model(s) | Physics inductive bias | Strength |
|---|---|---|
| UNet, UNet-CN | None | None |
| UNet-CP, UNet-CN-CP | Circular padding | Weak |
| FM | Continuous-time vector flow | Medium |
| FNO, TFNO, UFNO | Spectral layers | Strong |

Table 1: Overview of the eight models and the relative strength of their physics inductive biases.

prior consistent with periodic boundary conditions in the dataset. We also include our proposed FM model. Finally, we include three variants of the Fourier Neural Operator (FNO): a baseline model, a Tucker-factorized (TFNO) variant (Kossaifi et al., 2023) that improves parameter effiency, and a U-shaped (UFNO) variant (Rahman et al., 2023) designed to better capture multi-scale interactions.

**Training Procedure.** Across all models, we use a batch size of 16 and the AdamW optimizer (Loshchilov & Hutter, 2019) with $10^{-4}$ weight decay. For U-nets and FNOs, we adopt the coarse-tuned learning rates reported in *The Well* ($5 \cdot 10^{-4}$ and $10^{-3}$, respectively). For FM, we performed a separate coarse tuning and selected $5 \cdot 10^{-3}$. A linear warm-up cosine scheduler was applied for the first three epochs. The loss optimised is MSE averaged across fields and space. Training ran on a single Nvidia A40 GPU ($\leq$ 24h) with early stopping (patience 6). FM was trained in low-resolution data space with subsequent upsampling, avoiding the additional overhead of an autoencoder. Further details on FM in low-resolution data space are provided in App. E, and complete model configurations are listed in App. F.

### 4.4 ENERGY AND CARBON METRICS

We employ *Carbontracker* (Anthony et al., 2020) to estimate the energy consumption and carbon footprint during training and inference by monitoring real-time hardware power usage (CPU, GPU, DRAM) and adjusting for infrastructure overhead via the power usage effectiveness. The carbon footprint is reported as carbon dioxide equivalent ($CO_2$eq) emissions, which converts all greenhouse gases into an equivalent $CO_2$ warming potential. It is derived from the total energy consumed and by using the global average carbon intensity for 2024 at $445$ gCO$_2$/kWh, ensuring fair comparisons across models (IEA, 2025).

## 5 RESULTS

### 5.1 HARMONIC OSCILLATOR TOY EXPERIMENT

The results in Fig. 2 show that models with non-periodic activations perform poorly outside the training domain (extrapolation). Periodic activations (Sine, Snake) offer modest improvements, with the model using Snake activation function performing best. The PINN achieved superior results in both interpolation and extrapolation tasks; most notably for the damped harmonic oscillator shown in Fig 5. It remained stable and accurate even as prediction range increased and dynamics grew more complex.

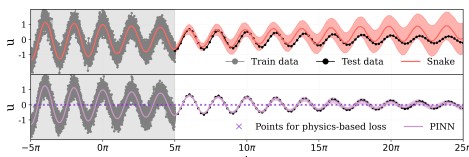

Figure 5: Damped harmonic oscillator predictions. Mean predictions (solid lines) with standard deviation (shaded areas) from 10 runs. Results are shown for the two best-performing models: Snake and PINN.

### 5.2 VISCOUS BURGERS' EQUATION EXPERIMENT

For the viscous Burgers' equation experiment, the comparison with NN, U-PINN, and S-PINN show interesting trends. Both PINNs achieve consistently lower test losses at lower carbon footprints compared to the supervised NN (see Fig. 1-b). Although the NN has the lowest carbon footprint for fixed epochs and data size, achieving a given accuracy is up to 10 times more carbon-efficient with

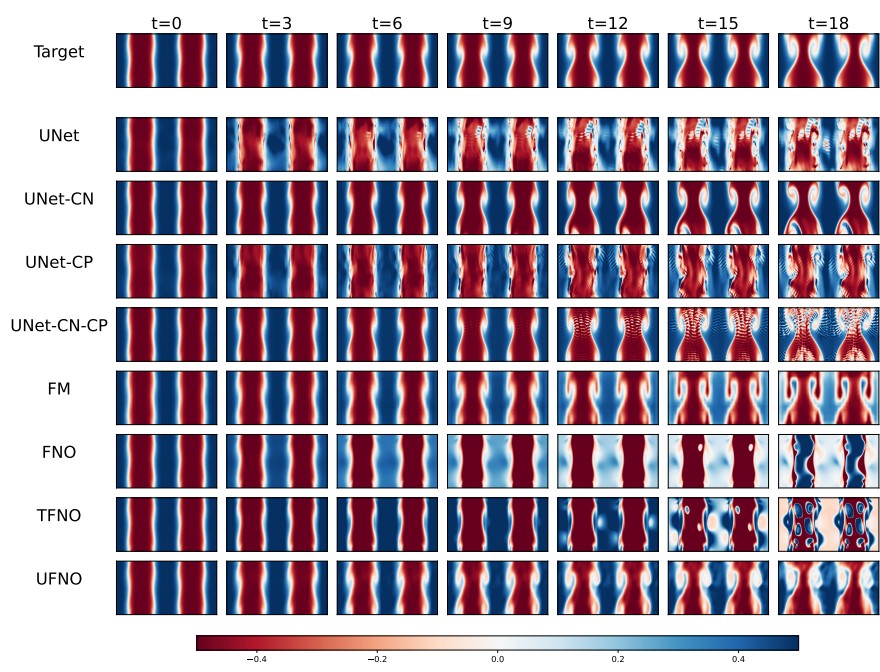

Figure 7: Example rollout predictions for one of the four fields (tracer). Target trajectory (top row) compared with model predictions shown every third step of a 20-step rollout, illustrating differences in long-horizon stability and fidelity. Models labels are ordered from no bias (top) to strong bias (bottom). Results for the three other fields are provided in Appendix G.

the PINNs. This S-PINN, which mixes data and physics, dominates the Pareto front in Fig. 1-b, suggesting that combining physics and data gives the model which balances efficacy and efficiency.

## 5.3 INCOMPRESSIBLE SHEAR FLOW EXPERIMENT

The first two simple experiments discussed above establish key insights into how physics can affect the model performance and resource consumption for specific tasks. In the incompressible shear flow experiment, we extend this analysis to a more complex task using a wide spectrum of models.

**Predictive Performance.** We first assess predictive performance using Pearson correlation $r$ at each step of a 20-step rollout, see Fig. 6. All eight models (see Table 1) achieve high accuracy for the first few steps ($r \approx 1$), after which performance declines at different rates. Over longer horizons, UNet-CN, UNet-CN-CP, and UFNO emerge as the strongest performers.

Figure 7 illustrates these stability differences visually. UNet and UNet-CP deteriorate rapidly, accumulating distortions within just a few steps. The modernised variants (UNet-CN, UNet-CN-CP) show marked improvements: UNet-CN captures fine-scale detail but gradually disrupts large-scale dynamics, while UNet-CN-CP better preserves both large and fine-scale structure at the cost of high-frequency artifacts. FM exhibits good temporal stability, though it tends to exaggerate some fine-scale structures. FNO and TFNO maintain the dominant large-scale dynamics but exhibit a systematic drift toward higher values, fail to capture fine-scale variability and eventually introduce distortions. UFNO better recovers both large- and small-scale structures, though these gradually smooth out over time.

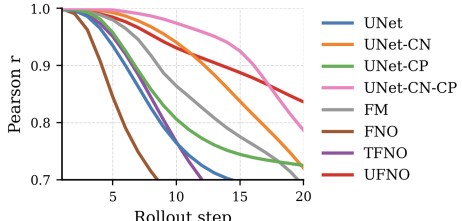

Figure 6: Pearson r over 20 rollout steps. Values are averaged over the four fields: velocity components, tracer, and pressure. Models labels are ordered from no bias (top) to strong bias (bottom).

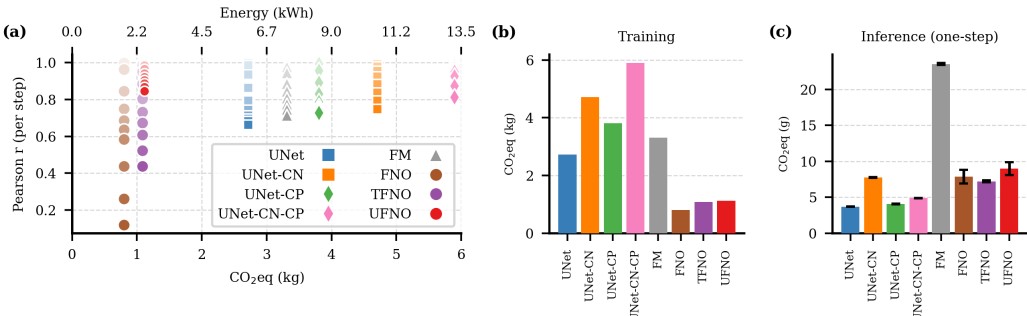

Figure 8: Predictive performance and carbon footprint. **a:** Predictive accuracy (Pearson r) versus training emissions (kgCO₂eq); each point represents every second step of a 20-step rollout (lighter shades indicate earlier steps), averaged over four fields. **b:** Training emissions (kgCO₂eq) from a single run per model. Models are ordered from no bias (left) to strong bias (right). **c:** One-step inference emissions (gCO₂eq) from three repeated runs over the full test set; error bars indicate variability across repetitions. Note that inference costs also include the initial model loading cost.

**Training Cost.** We next relate predictive performance to training carbon footprint (Fig. 8-a), revealing pronounced differences across models. FNO variants are the most efficient, with carbon footprints 2.5–7.5× smaller than those of U-net variants and FM. The top-performing models, UNet-CN, UNet-CN-CP, and UFNO reveal very different training costs: the training carbon footprint of UFNO is less than one-fourth and one-fifth of UNet-CN's and UNet-CN-CP's, respectively.

**Inference Cost.** To provide a more complete evaluation, we also account for inference costs of all the trained models. This can be useful in choosing spatio-temporal forecasting models as inference costs can dominate over training or development costs.

Figure 8-b and c contrast training emissions with one-step inference emissions. While the FNO variants are the most carbon-efficient during training, the picture shifts at inference. Here, the FNO variants are on average more expensive than the U-net variants. FM stands out, with inference costs 2–5.5× higher than all other models. These discrepancies emphasise the importance of considering both training and inference rather than focusing on a single stage.

Padding choices further illustrate the sensitivity of emissions to architectural design. Switching from zero to circular padding increases training emissions for both UNet and UNet-CN. At inference, the effects diverge: circular padding slightly increases the footprint for UNet, while significantly reducing it for UNet-CN. This highlights how even minor inductive design choices – in this case enforcing circular padding to reflect periodic boundaries – can shift the efficacy–efficiency balance.

Table 6 in App. G reports both training and inference footprints alongside run times. While runtime and energy consumption/ carbon emissions are closely linked, they are not strictly proportional, as some operations could take longer but use less energy, and vice-versa (Henderson et al., 2022; Bakhtiarifard et al., 2024). Additional results for these models can be found in App. G.

## 6 DISCUSSIONS

Our study demonstrates that physics inductive biases impact the trade-off between efficacy and efficiency. In simple settings, such as the harmonic oscillator, adding periodic structure or a physics-informed loss enables extrapolation beyond the training range. With viscous Burgers' equation, combining data with physics yields the model that dominates the Pareto front, achieving lower test errors at lower carbon cost compared to the purely data-driven baseline.

In the incompressible shear flow setting, we observe clear differences in how the physics priors affect both predictive performance and carbon footprint across training and inference. Models with strong physics inductive biases achieve the lowest training footprints, but this comes at the expense of higher inference costs compared to U-net variants. UFNO achieves one of the most favourable overall profiles, combining strong predictive accuracy with a low training footprint. Its inference cost, however, remains nearly twice that of the best-performing U-Net variant (UNet-CN-CP). The

results underscore the importance of evaluating training and inference together. A model that trains cheaply may prove inefficient when repeatedly deployed, while others entail higher upfront cost but amortise more effectively in deployment. Amortising development costs across the model lifecycle is one way to factor in these considerations (Cottier et al., 2024).

FM stands out as an outlier: while competitive in predictive performance, its reliance on numerical integration makes inference substantially more expensive. This is in spite of it operating on downsampled data, as the original resolution was computationally prohibitive (Dao et al., 2023).

Several key points follow from these results. First, model selection cannot be based on efficacy alone, as models with similar predictive accuracy can differ substantially in carbon emissions. Second, once both training and inference costs are accounted for, identifying a single "best" model becomes less straightforward, as illustrated by the incompressible shear flow experiment. A model with a small carbon footprint during training may be unsuitable in deployment, making carbon-aware evaluation across the full lifecycle essential.

The implications extend well beyond our test-bed. Many scientific and engineering problems are governed by well-understood structures, and overlooking them is wasteful. At the same time, tailoring models too closely to the specific physics can limit applicability and can be challenging for complex systems. This motivates the development of general-purpose models such as FNOs and FM, which allow physics priors to be incorporated more easily while retaining flexibility across tasks. However, as demonstrated by FM, alignment with the underlying physics does not guarantee carbon efficiency. Physics should not be incorporated at any cost, but only where it provides clear benefits in terms of predictive performance or carbon efficiency. While our discussion centres on physics-inductive biases, the same logic extends to other forms of prior knowledge: any regularising bias that constrains the learning task appropriately can potentially offer similar trade-offs between efficacy and efficiency.

Ultimately, our results argue for a shift in mindset. Progress in ML should not be equated with ever-larger datasets, models, and compute budgets, but rather evaluated in terms of the value gained relative to the cost incurred. Learning everything from data while ignoring domain knowledge reflects an abundance mindset, in which resources are treated as unlimited. A more sustainable alternative is to exploit what we already know by embedding this knowledge into models. Trading carbon for physics offers a principled path towards architectures that leverage prior knowledge, reduce reliance on brute-force compute, and achieve more favourable efficacy-efficiency trade-offs.

**Limitations.** This study has some limitations. First, all models were trained without extensive hyperparameter tuning. This choice emphasises architectural comparison over peak performance, trading statistical robustness for a broader perspective on performance–carbon trade-offs. In practice, however, hyperparameter search can dominate computational cost, and complex models typically require many more training runs during development than simpler baselines. Second, we restricted our study to spatio-temporal forecasting. While this limits the scope of the contribution, the central message about taking efficiency considerations into account by including appropriate inductive biases is applicable across ML tasks. Finally, our measurements do not account for hardware specific optimisations, where certain operations are more efficiently executed on specific accelerators. This, for instance, could yield an advantage to deep learning models compared to PDE solvers.

## 7 CONCLUSIONS

The primary message in this work, backed by empirical evidence, is to increase resource-awareness when designing ML experiments. While this can be accomplished in several ways (Bartoldson et al., 2023), we focus on the role of physics inductive biases and illustrate the different decisions practitioners have to consider. Through multiple experiments from simple oscillators to complex shear flow, we showed how physics-inductive biases affect the trade-off between predictive performance and carbon footprint.

This work advances a perspective grounded in resource-awareness and calls for a shift in mindset. As a community, we should discourage the pursuit of marginal improvements in performance that come from disproportionality higher resource costs. With our work we have made the case to encourage resource-awareness by utilising existing domain knowledge and/or other inductive biases.

ACKNOWLEDGMENTS

Authors thank several people.

ETHICS STATEMENT

This work advocates for resource-awareness when developing and using ML. We discuss the carbon footprint of ML and the implications of pursuing *abundance mindset*. We consider this to be contributing to a more ethical discourse around ML. We do not foresee any negative ethical implications of this work.

REPRODUCIBILITY STATEMENT

The dataset and code used in the viscous Burgers' equation experiment is from Khoa (2022). For the main experiment, the dataset and code for FNO, TFNO, UNet, and UNet-CN are from Ohana et al. (2024) with identical model configurations. UNet variants with CP are implemented by modifying the padding scheme. UFNO is implemented using the same operator learning library as the other FNO variants (Kossaifi et al., 2025), following the configuration in Rahman et al. (2023). The flow matching model builds on Lim et al. (2025) but is adapted to operate directly in data space using the implementation at Anonymous (2025).

USE OF LARGE LANGUAGE MODELS

We have made use of public available large language models for programming assistance and for refining the writing.

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

APPENDIX

## A FLOW MATCHING

We condition on two consecutive states $(u_0, u_1) := (u_k, u_{k+1})$ such that the path interpolates between current state $u_0$ and the target state $u_1$. Following Lim et al. (2025), we adopt their interpolation path:

$$p_\tau(u \mid u_0, u_1) = \mathcal{N}\big(u \mid (1-\tau)u_0 + \tau u_1, \ (\sigma_{\min}^2 + \sigma^2\tau(1-\tau))I\big), \tag{2}$$

with $\sigma_{\min} = 10^{-3}$ and $\sigma = 10^{-1}$, as recommended for Navier–Stokes data. The mean interpolates linearly between $u_0$ and $u_1$, while the variance follows a Brownian-bridge profile (maximal halfway at $\tau = 0.5$, minimal at endpoints at $\tau = 0$ and $\tau = 1$).

The corresponding vector field is:

$$v_\tau(u \mid u_0, u_1) = u_1 - u_0 + \frac{\sigma^2}{2} \frac{1 - 2\tau}{\sigma_{\min}^2 + \sigma^2\tau(1-\tau)} \Big(u - \big((1-\tau)u_0 + \tau u_1\big)\Big). \tag{3}$$

The parametric vector field $v_\tau^\theta$ is trained to approximate $v_\tau$ by minimizing:

$$\mathcal{L}(\theta) = \mathbb{E}_{\tau, \ p_\tau(u|u_0,u_1), \ q(u_0,u_1)} \Big\| v_\tau^\theta(u) - v_\tau(u \mid u_0, u_1) \Big\|^2. \tag{4}$$

Here $\tau \sim \mathcal{U}[0,1]$, $u \sim p_\tau(u \mid u_0, u_1)$, and $(u_0, u_1) \sim q(u_0, u_1)$, where $q$ denotes the empirical distribution of consecutive state pairs in the dataset.

In addition to the conditioning pair $(u_0, u_1)$, we provide temporal context by sampling one additional state $u_c \in \{u_{k-3}, u_{k-2}, u_{k-1}\}$, giving a maximum horizon $h = 4$ consistent with FNOs and U-nets. All inputs are downsampled by a factor of 16 via average pooling, and predictions are upsampled to the original resolution with bilinear interpolation.

During inference, given a state $u_k$, the learned flow is simulated by numerically integrating $v_\tau^\theta$ from $\tau = 0$ to $\tau = 1$. We use Euler's method with 10 integration steps, which produces intermediate states approximating the continuous trajectory. The final iterate is then upsampled to the original resolution, yielding the forecast $\hat{u}_{k+1}$.

## B TOY EXPERIMENT ON HARMONIC OSCILLATOR

We design a toy experiment to study the effect of physics inductive biases for extrapolation in dynamical systems. The task is to model the trajectory of a harmonic oscillator, comparing standard feedforward networks with variants that incorporate different inductive biases. Specifically, we evaluate four networks with identical architecture but different activation functions, `ReLU`, `Tanh`, `Sine`, and `Snake` (Ziyin et al., 2020), where:

$$\text{Sine}(x) = \sin(x), \qquad \text{Snake}(x) = x + \sin^2(x). \tag{5}$$

The `Sine` and `Snake` activations introduce oscillatory biases aligned with the dynamics of the oscillator. In addition, we include a PINN that augments the data loss with a PDE residual while using `Snake` activations.

The governing dynamics are given by the damped harmonic oscillator:

$$m\frac{d^2u}{dt^2} + \mu\frac{du}{dt} + ku = 0, \tag{6}$$

$$u(0) = 1, \ \text{and} \ \frac{du}{dt} = 0. \tag{7}$$

The damping ratio is defined as $\delta = \mu/(2m)$. We adopt non-dimensionalized units with $m = 1$ and $\omega_0 = \sqrt{k/m} = 1$. We consider both the undamped case ($\delta = 0$) and a damped case ($\delta = 0.02$).

Training data are sampled from $t \in [-5\pi, 5\pi]$ with Gaussian noise ($\mu = 0, \sigma = 0.3$). The purely supervised networks use $10,000$ supervised samples, while the PINN uses 100 points (1%) for the

physics loss. Test data are noise-free and sampled over $t \in [5\pi, 25\pi]$ to assess extrapolation. The points for evaluating the physics-based loss are sampled across the entire domain.

The purely supervised networks minimize the MSE:

$$\mathcal{L}_{\text{NN}} = \mathcal{L}_{\text{data}} = \frac{1}{N_{\text{data}}} \sum_{i=1}^{N_{\text{data}}} (u_i - \hat{u}_i)^2, \tag{8}$$

where $u$ denotes the ground truth and $\hat{u}$ the model predictions.

The PINN combines the same data term with a differential equation (DE) residual:

$$\mathcal{L}_{\text{PINN}} = \mathcal{L}_{\text{data}} + \mathcal{L}_{\text{pde}} = \frac{1}{N_{\text{data}}} \sum_{i=1}^{N_{\text{data}}} (u_i - \hat{u}_i)^2 + \frac{1}{N_{\text{de}}} \sum_{j=1}^{N_{\text{de}}} \left( \frac{d^2 \hat{u}_j}{dt^2} + \mu \frac{d\hat{u}_j}{dt} + k \hat{u}_j \right)^2. \tag{9}$$

Further architectural and optimisation details are given in Sec. 4.1.

## C  Loss Functions Used in the viscous Burgers' Equation Experiment

The loss function for the purely data-driven NN, denoted $\mathcal{L}_{\text{NN}}$, is the MSE loss given by:

$$\mathcal{L}_{\text{NN}} = \mathcal{L}_{\text{data}} = \frac{1}{N_{\text{data}}} \sum_{i=1}^{N_{\text{data}}} (u_i - \hat{u}_i)^2. \tag{10}$$

Here $u$ and $\hat{u}$ denote the ground truth and the predictions.

The U-PINN optimizes solely using the governing equations including those that describe the boundary and initial conditions to ensure a unique solution. Its loss function, $\mathcal{L}_{\text{U-PINN}}$, is composed of the residuals of the PDE and the deviations from the initial and boundary conditions, expressed as:

$$\mathcal{L}_{\text{U-PINN}} = \mathcal{L}_{\text{pde}} + \mathcal{L}_{\text{ic}} + \mathcal{L}_{\text{bc}} \tag{11}$$

$$= \frac{1}{N_{\text{pde}}} \sum_{i=1}^{N_{\text{pde}}} \left( \frac{\partial \hat{u}_i}{\partial t} + \hat{u}_i \frac{\partial \hat{u}_i}{\partial x} - \nu \frac{\partial^2 \hat{u}_i}{\partial x^2} \right)^2 \tag{12}$$

$$+ \frac{1}{N_{\text{ic}}} \sum_{j=1}^{N_{\text{ic}}} \left( \hat{u}(x_j, 0) + \sin(\pi x_j) \right)^2$$

$$+ \frac{1}{N_{\text{bc}}} \sum_{k=1}^{N_{\text{bc}}} \left( \hat{u}(-1, t_k)^2 + \hat{u}(1, t_k)^2 \right).$$

The S-PINN combines both data and physics losses. Its total loss function, $\mathcal{L}_{\text{S-PINN}}$ is given by:

$$\mathcal{L}_{\text{S-PINN}} = \mathcal{L}_{\text{NN}} + \mathcal{L}_{\text{U-PINN}} = \mathcal{L}_{\text{data}} + \mathcal{L}_{\text{pde}} + \mathcal{L}_{\text{ic}} + \mathcal{L}_{\text{bc}}. \tag{13}$$

## D  Incompressible Shear Flow Dataset

This experiment is based on data from *The Well* (Ohana et al., 2024), consisting of 2D incompressible Navier–Stokes dynamics with an additional passive tracer:

$$\partial_t \mathbf{u} + \nabla p - \nu \nabla^2 \mathbf{u} = -\mathbf{u} \cdot \nabla \mathbf{u}, \tag{14}$$

$$\partial_t s - D \nabla^2 s = -\mathbf{u} \cdot \nabla s, \tag{15}$$

$$\nabla \cdot \mathbf{u} = 0. \tag{16}$$

Here $\mathbf{u} = (u_x, u_y)$ is the velocity components, $p$ is pressure satisfying $\int p = 0$, $s$ is the tracer, $\nu$ is viscosity, and $D$ is diffusivity. The PDE parameters are linked to Reynolds and Schmidt numbers as $\nu = 1/\text{Re}$ and $D = \nu/\text{Sc}$.

The system is initialized with a horizontal shear flow:

$$u_x(y) = \tanh\left(5\frac{y - y_k}{n_{\text{shear}}w}\right), \tag{17}$$

where $y_k$ is the vertical position of the $k$-th shear layer, $n_{\text{shear}}$ is the number of shear layers, and $w$ is a width parameter that controls how sharply the velocity changes across each layer.

To introduce small perturbations in the system, the vertical velocity field $v_y$ contains sinusoidal variations along the $x$-direction, localized at the shear layers. These perturbations are given by:

$$u_y(x, y) = \sin(n_{\text{blobs}}\pi x)\exp\left(\frac{-25}{w^2}|y - y_k|^2\right). \tag{18}$$

The number of oscillations is controlled by $n_{\text{blobs}}$, while $w$ influences how localized these perturbations are around the shear layers.

The tracer field is initialized to match the shear flow, while the pressure field is initialized to zero everywhere in the domain. Thus, the initial conditions are fully characterized by three parameters: $n_{\text{shear}} \in \{2, 4\}$, $n_{\text{blobs}} \in \{2, 3, 4, 5\}$, and $w \in \{0.25, 0.5, 1.0, 2.0, 4.0\}$. An example of the data is illustrated in Fig. 9.

As part of preprocessing, we standardize each field using $\hat{x} = (x - \mu)/\sigma$ (see Table 2). This normalization balances the contribution of different fields by preventing those with high variance from dominating the loss.

Table 2: Field statistics. Mean and standard deviation of raw fields, averaged over all time steps.

| Field | Mean | Std |
|---|---|---|
| s | $2.19 \times 10^{-3}$ | 0.33 |
| p | $-1.18 \times 10^{-9}$ | 0.074 |
| $v_x$ | $2.19 \times 10^{-3}$ | 0.40 |
| $v_y$ | $-1.40 \times 10^{-7}$ | 0.097 |

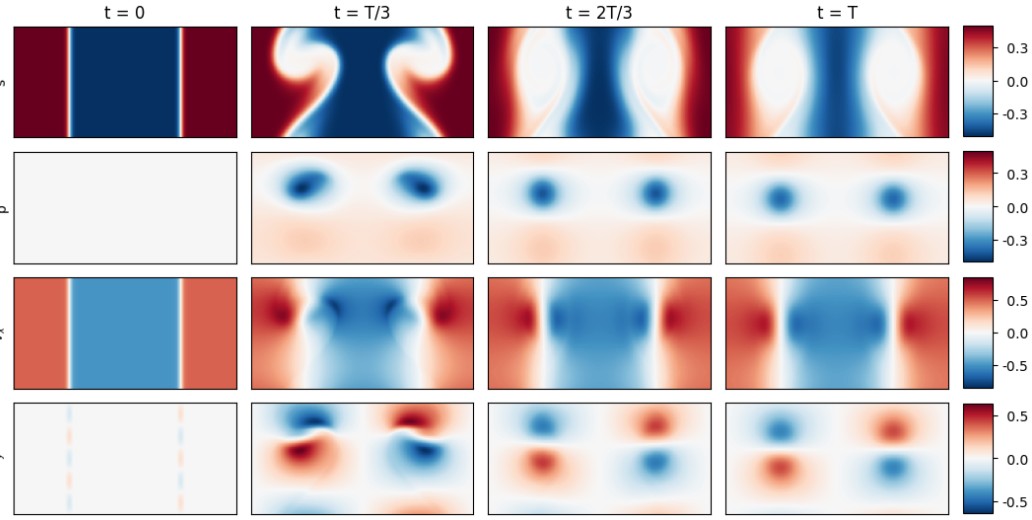

Figure 9: Example of incompressible shear flow data. Temporal evolution of tracer (s), pressure (p), horizontal velocity ($v_x$) and vertical velocity ($v_y$) fields at four time steps: $t \in \{0, T/3, 2T/3, T\}$. The snapshots illustrate the coupled dynamics of the fields.

## E    FLOW MATCHING IN LOW-RESOLUTION DATA SPACE

Most prior work applies FM in latent space, with a few exceptions (Disch et al., 2025; Baldan et al., 2025) demonstrating its feasibility directly in data space. We adopt this latter strategy to avoid the additional energy and carbon costs of an encoder–decoder pipeline. To make FM computationally tractable, we downsample inputs by a factor of 16, perform training and inference at low resolution, and subsequently upsample to the original resolution. This design inevitably encode a trade-off

between efficacy and efficiency: while operating in low-resolution data space suppresses fine-scale dynamics and may introduce artifacts, it delivers substantial efficiency gains, reducing training and inference time by orders of magnitude.

## F   MODEL CONFIGURATIONS

Model configurations are adapted from prior work. U-nets, FNO, and TFNO follow Ohana et al. (2024), UFNO follows Rahman et al. (2023), and FM is adapted from Lim et al. (2025), with some adjustments to match our setting (including operating directly in data space). In line with *The Well* (Ohana et al., 2024), U-nets and FNO variants are scaled to $\sim$17–20M parameters, while FM is smaller ($\sim$8.5M). Table 3 provides an overview of parameter counts, with detailed configurations summarized in Table 4 (U-nets and FNOs) and Table 5 (FM).

Table 3: Model parameters.

| Model | #Params |
|---|---|
| UNet | 17.5M |
| UNet-CN | 18.6M |
| UNet-CP | 17.5M |
| UNet-CN-CP | 18.6M |
| FM | 8.6M |
| FNO | 19.0M |
| TFNO | 19.3M |
| UFNO | 18.8M |

Table 4: U-nets and FNOs model configurations.

| Model | Filter size/modes | Init dim | Blocks/stage | Up/down | Bottleneck |
|---|---|---|---|---|---|
| UNet, UNet-CP | 3 | 48 | 1 | 4 | 1 |
| UNet-CN, UNet-CN-CP | 7 | 42 | 2 | 4 | 1 |
| FNO, TFNO | 16 | 128 | 4 | – | – |
| UFNO | 8, 12, 16 | 128 | 1 | 3 | 1 |

Table 5: FM model configuration. The FM model operates in low-resolution data space and is built from transformer stacks with learned positional, timestamp, and distance embeddings.

| Model | Resolution | Init dim | Depth | Bottleneck | Heads |
|---|---|---|---|---|---|
| FM | $16 \times 32$ | 256 | 3 | 4 | 8 |

## G   ADDITIONAL RESULTS

This appendix presents supplementary plots that support the findings in the main text. Table 6 reports both the training and inference emissions alongside run times.

| Model | Training | | Inference (one-step) | |
|---|---|---|---|---|
| | $CO_2$eq (kg) | Time (h) | $CO_2$eq (g) | Time (min) |
| UNet | 2.72 | 12.7 | $3.68 \pm 0.03$ | 1.0 |
| UNet-CN | 4.71 | 15.9 | $7.77 \pm 0.05$ | 1.9 |
| UNet-CP | 3.81 | 14.2 | $4.06 \pm 0.05$ | 1.0 |
| UNet-CN-CP | 5.89 | 20.9 | $4.87 \pm 0.02$ | 1.2 |
| FM | 3.31 | 14.0 | $23.54 \pm 0.13$ | 4.6 |
| FNO | 0.80 | 4.0 | $7.87 \pm 0.94$ | 2.0 |
| TFNO | 1.09 | 5.3 | $7.21 \pm 0.12$ | 1.8 |
| UFNO | 1.12 | 3.6 | $9.00 \pm 0.88$ | 2.4 |

Table 6: Carbon footprint during training and inference. Reported $CO_2$eq emissions and runtime for all models, measured with *CarbonTracker*. Inference results correspond to one-step forecasts over the full test set (4,704 samples) and includes the initial model loading cost. Each inference experiment was repeated three times; $CO_2$eq is reported as mean and standard deviation, and runtimes as mean values.

Fig.10-a reports model performance using the variance-scaled root mean squared error (VRMSE), which normalises prediction error by the variance of the ground truth to account for scale differences across fields:

$$\text{VRMSE}(u, \hat{u}) = \left( \frac{\langle |u - \hat{u}|^2 \rangle}{\langle |u - \langle u \rangle|^2 \rangle + \epsilon} \right)^{1/2}, \quad (19)$$

where $\langle \cdot \rangle$ denotes spatial averaging, $u$ the ground truth, $\hat{u}$ the prediction, and $\epsilon = 10^{-8}$ a small constant for stability. The VRMSE measures errors relative to the natural variability of the data, with $\text{VRMSE}(y, \langle y \rangle) \approx 1$ implying that values greater than one indicate prediction errors larger than typical fluctuations. The results follow the same overall trends as in Fig.8-a, where performance was measured with the Pearson correlation coefficient.

Figure 10-b reports the inference carbon footprint for 24 autoregressive rollouts of 50 steps each. Compared to one-step inference over the full test set (4,704 samples), the rollout costs tend to average out across models, since the fixed overhead of model loading constitutes a larger share of the total cost. Flow matching remains an outlier, standing out as the most expensive model during inference.

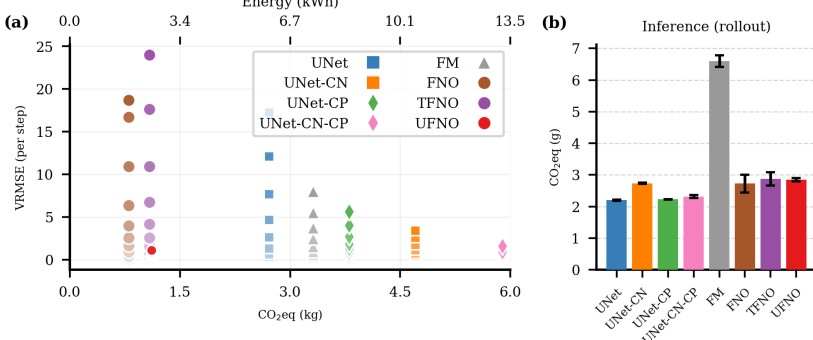

Figure 10: Predictive performance and rollout carbon footprint. **a:** VRMSE vs. training carbon footprint. Each point shows every second step of a 20-step rollout, averaged across the four fields (lighter shades indicate earlier steps). **b:** Carbon footprint of inference during rollouts. Reported $CO_2$eq emissions for 24 autoregressive rollouts of 50 steps. Inference costs include the initial model loading cost. Models are ordered from no bias (left) to strong bias (right).

Figs. 11, 12, and 13 provide rollout prediction examples for the three physical fields not shown in the main text (pressure fields and the two velocity components). The observations made for the tracer field in the main text extend to these additional fields.

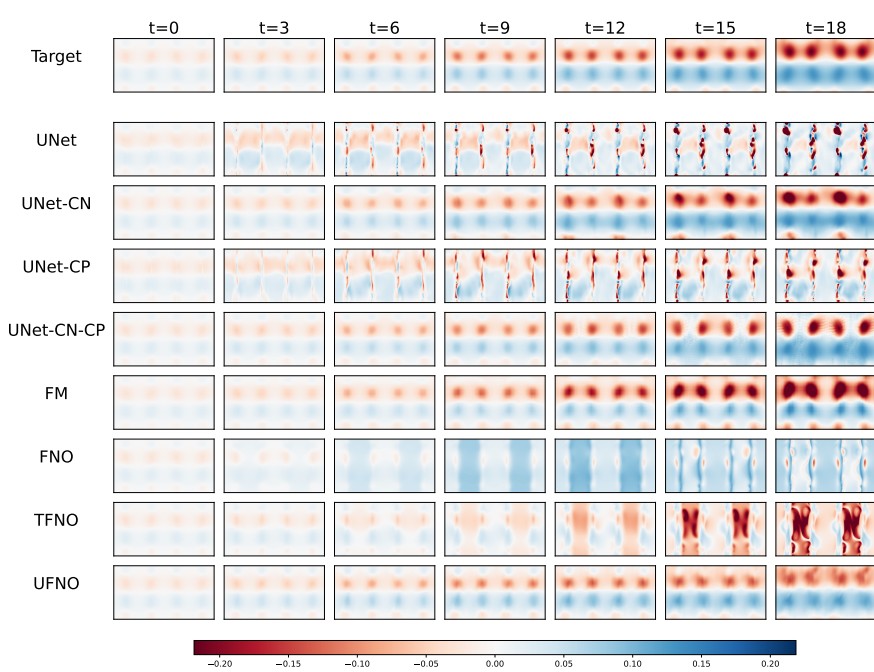

Figure 11: Example of rollout predictions for pressure field. Target trajectory (top row) compared with model predictions shown every third step of a 20-step rollout. Models labels are ordered from no bias (top) to strong bias (bottom)

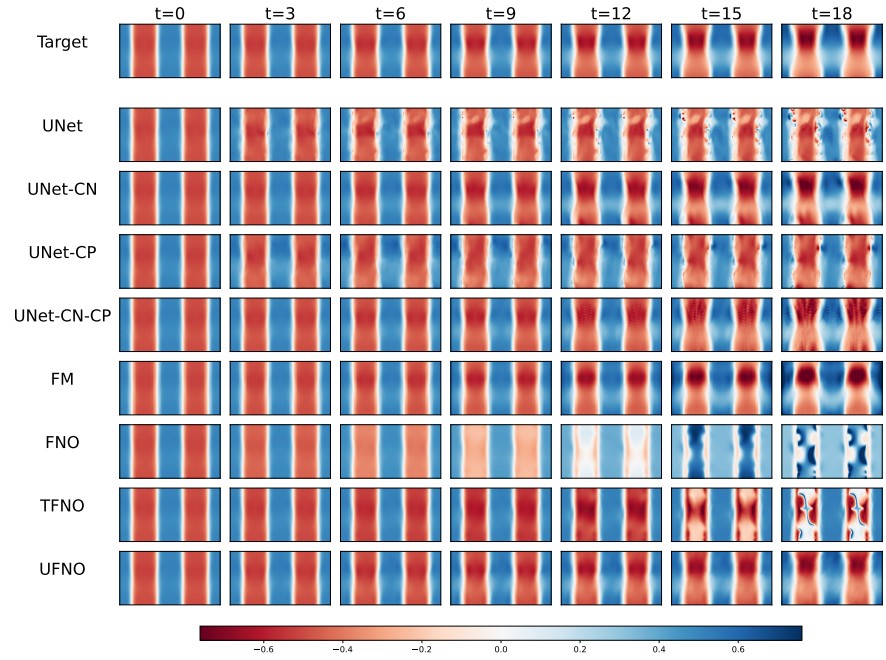

Figure 12: Example of rollout predictions for horizontal velocity field. Target trajectory (top row) compared with model predictions shown every third step of a 20-step rollout. Models labels are ordered from no bias (top) to strong bias (bottom)

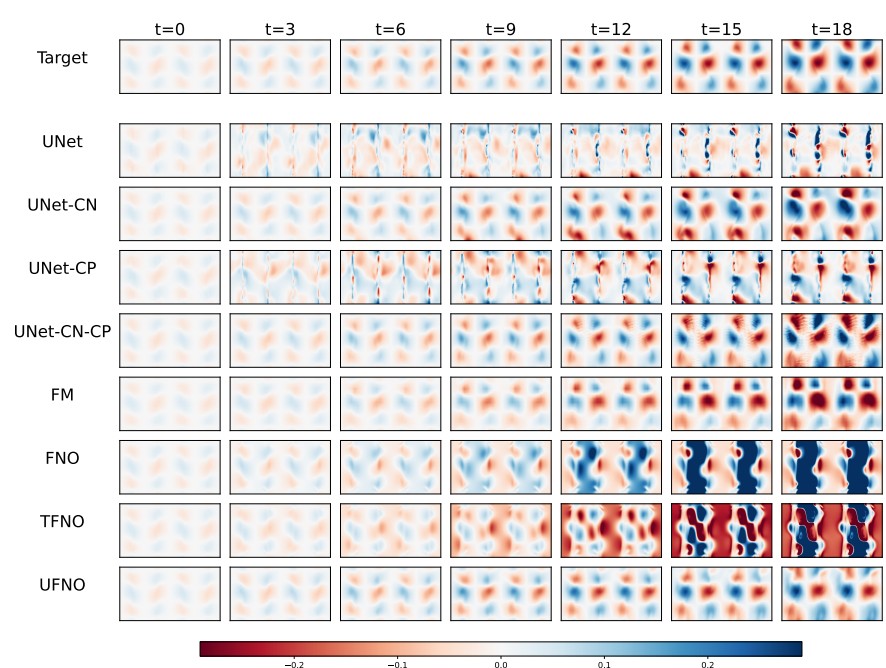

Figure 13: Example of rollout predictions for vertical velocity field. Target trajectory (top row) compared with model predictions shown every third step of a 20-step rollout. Models labels are ordered from no bias (top) to strong bias (bottom)

