# OpenReview forum: "Trading Carbon for Physics:  On the Resource Efficiency of Machine Learning for Spatio-Temporal Forecasting"
_ICLR.cc/2026/Conference — Submitted to ICLR 2026_

### Official Review · Reviewer_Y12R · 2025-10-15

**Soundness:** 3
**Presentation:** 3
**Contribution:** 2
**Rating:** 2
**Confidence:** 3

**Summary:**

This paper explores the paradigm of Physics informed Machine Learning in the context of tradeoff between efficiency and accuracy. The authors deep dive into how introducing physics biases into pure data-driven methods can reduce carbon footprint of the computational heavy deep learning models. The authors provide in-depth empirical results for the simplest physics problem (oscillation) to more complex physical phenomenon (navier-stokes). The experimental setup shows during training pure data-driven methods like UNet variants perform better in terms of accuracy and strong physics methods like FNO perform better in terms of efficiency with lowest carbon footprint. However, during inference FNO variants and Flow Matching models costs higher than pure data-driven models.

**Strengths:**

1. The paper explores an important subject of carbon footprint of pure data-driven methods (deep learning models).
2. In-depth experiments are performed targeting simplest physics to more complex physics phenomena.
3. The paper presents both efficacy and efficiency metrics at both training and inference times.
4. The discussion section of the paper is clearly and well-written highlighting how physics alone is not the answer of reducing carbon footprint.

**Weaknesses:**

1. The authors emphasise on the trade-off for carbon footprint and accuracy, however as the problem gets complex the trade-off seems to disappear as Unet based variants do perform better at inference times as well as have a lower carbon footprint, while FM which is supposed to be a mid point for physics and data-driven has the highest carbon footprint at inference time.
2. The study is solely done on synthetic dataset, it would be good to see some examples, efficacy and efficiency metrics for some real world datasets.
3. In my point of view the study is largely experimental and I couldn't pinpoint the innovative aspect of it. Therefore, I think it needs to be backed up by a lot more experiments convincing that there indeed exists a accuracy–carbon trade-off in introducing strong and weak physics inductive bias.

**Questions:**

1. An experimental setup on real world dataset would definitely strengthen the study.
2. Additional experiments on real world dataset exploring the costs at training and inference times
3. More insights into why the carbon footprint landscape changes during training and inference time. for example can authors provide an insight into why FM inference cost is so high, does it solely have to do with the increase in roll-out steps?
4. Figure 8 a shows the predictive performance vs carbon footprint at training time, could authors include the same figure or results for inference time as well for longer rollouts?

---

> ### Author Response · Authors · 2025-11-18
>
> We thank the reviewer for their time and appreciate the positive remarks on the importance of the work, the depth of the experiments, and the quality of the discussion.
>
> We would like to address the points and questions they make in their review:
>
> **W1. Trade-off becomes less straightforward as problem complexity increases**
>
> We acknowledge that the efficacy–efficiency trade-off becomes less clear as problem complexity increases, as discussed in L440–445. This reinforces the need for multi-objective evaluation that incorporates energy and carbon considerations, since selecting the best model is no longer straightforward.
>
> **W2. Synthetic datasets**
>
> We agree that applying the analysis to real-world datasets would be valuable, but this is beyond the scope of the present work.
>
> **W3. Experimental contribution**
>
> We acknowledge that the contribution is primarily experimental. Our intention is to demonstrate the efficacy–efficiency trade-off using a controlled set of studies while keeping resource use modest, which is why we include two small-scale experiments alongside one large-scale case. We believe the current evidence is sufficient to support the main message and therefore do not plan additional experiments (also see the general response to all reviewers).
>
> **Q1+Q2. Additional experiments on real world datasets**
>
> We agree this would strengthen the study, but it is beyond the scope of the present work.
>
> **Q3. Insights into differences in carbon footprint at training and inference**
>
> The FM model requires numerical integration at inference, which introduces multiple forward passes per prediction. This makes inference inherently more expensive than single-pass models such as U-Nets or FNOs. We will clarify this in the paper.
>
> **Q4. Additional figures**
>
> We appreciate the suggestions. However, inference time is highly correlated with the carbon cost shown in Figure 8c, so an additional plot would not provide further insights. Moreover, computing the accuracy for longer rollouts are not meaningful, as predictions have degraded substantially by around twenty steps (as seen in Figure 7), making further comparison unreliable.

---

### Official Review · Reviewer_JQDf · 2025-10-31

**Soundness:** 2
**Presentation:** 2
**Contribution:** 2
**Rating:** 4
**Confidence:** 3

**Summary:**

This paper advocates for elevating resource efficiency (e.g., carbon footprint) to a primary consideration alongside efficacy in machine learning model development. Through a series of experiments on spatio-temporal forecasting tasks, the work systematically investigates how physics-inductive biases affect the trade-off between model performance and carbon cost. The authors compare a spectrum of models, from purely data-driven U-Nets to models with strong physical priors like PINNs and Fourier Neural Operators (FNOs). The study finds that incorporating physics biases generally leads to significant reductions in training costs, though this advantage may diminish or even reverse during inference.

**Strengths:**

- Important Topic: The paper tackles the sustainability of machine learning, a topic of growing importance and immense practical relevance to the community. It makes a strong case for shifting from a performance-only paradigm to one that balances efficacy and efficiency.
- Comprehensive Empirical Study: The systematic comparison across a spectrum of models with varying degrees of physics bias is rare and valuable. The detailed carbon footprint data provides an excellent baseline for future researchers.
- Nuanced Analysis: The paper's sharp distinction between training and inference costs is an often-overlooked but critical detail. This finding deepens the understanding of a model's lifecycle cost.

**Weaknesses:**

- Limited Novelty: The main weakness is the originality of the central thesis. The paper validates and quantifies what is largely "common knowledge" in the Scientific ML community: that physics constraints can reduce the reliance on brute-force computation. While the validation itself is valuable, it does not constitute a fundamental new discovery or method.
- Lack of Hyperparameter Tuning: The authors acknowledge that all models were trained without extensive hyperparameter optimization. This is a significant limitation. Different architectures may have different sensitivities to hyperparameters, and the reported trade-offs might be artifacts of specific (and potentially suboptimal) configurations rather than intrinsic properties of the model families. This could affect the robustness of the observed Pareto fronts.
- Generalizability of Conclusions: The study is focused exclusively on PDE-based spatio-temporal forecasting. While an excellent testbed, the paper does not deeply explore the generalizability of its findings to other domains where inductive biases are critical (e.g., equivariance in computer vision, structural biases in NLP).

**Questions:**

- How do the authors distinguish their contribution from the established consensus in the Scientific ML community that physics biases improve efficiency? Beyond providing quantitative evidence, does this work offer a new conceptual understanding?
- The lack of hyperparameter tuning could systematically affect the results. For example, is it possible that a more complex architecture like a U-Net could benefit disproportionately more from tuning than an FNO, potentially altering the cost-performance trade-offs?
- The Flow Matching model stands out as an outlier with high inference costs. Could the authors elaborate on the fundamental algorithmic or architectural reasons for this? Is this inefficiency inherent to the method or an artifact of the specific implementation used?

---

> ### Author Response · Authors · 2025-11-18
>
> We thank the reviewer for their time and appreciate the positive remarks regarding the importance of the topic, the comprehensive empirical study, and the nuanced discussion. We address the reviewer’s points and questions below.
>
> **W1. Limited novelty**
>
> We agree that the paper does not introduce new methods. As the reviewer notes, its contribution lies in providing a comprehensive empirical study of how physics inductive biases affect the carbon footprint of data-driven models. The main aim is to promote resource-aware practice and encourage a shift away from pursuing marginal accuracy gains at disproportionate computational and carbon costs.
>
> **W2. Hyperparameter tuning**
>
> We should clarify this point as the current wording in the limitations section understates what was actually done. As detailed in Appendix F, six of the eight models use the hyperparameters reported in [1], which were tuned on several datasets including the one used here. UFNO follows the configuration from [2], which is the only model not tuned specifically for this dataset. For FM, we performed additional hyperparameter calibration for this task. We will revise the limitations section accordingly:
>
> Original:
> > First, all models were trained without extensive hyperparameter tuning.
>
> Edited:
> > First, we do not perform full hyperparameter sweeps for every model. As detailed in Appendix F, most configurations follow tuned settings from prior work on several datasets, including the shear-flow dataset used here.
>
> [1] Ohana et al. The well: A large-scale collection of diverse physics simulations for machine learning. Dataset, 2024. URL https://doi.org/10.17863/CAM.113689.
>
> [2] Rahman et al. U-no: U-shaped neural operators. Transactions on Machine Learning Research, 2023. URL https://doi.org/10.48550/arXiv.2204.11127.
>
>
> **W3. Generalisability of conclusions**
>
> We agree that our study focuses on PDE-based spatio-temporal forecasting and as the reviewer rightly points out it is primarily serving as a test-bed; which we use to study more explicit forms of inductive biases. In presenting the related works, we already identify existing works which explore other methods that identify reduction of training compute, data or accelerate convergence in other classes of problems (L155-158). These methods, however, do not explicitly study the efficacy-efficiency trade-off from an energy consumption or carbon footprint point of view. We will also add a discussion point to clarify our contribution by elaborating on the existing point (L452):
>
> Original:
> >  While our discussion centres on physics-inductive biases, the same logic extends to other forms of prior knowledge: any regularising bias that constrains the learning task appropriately can potentially offer similar trade-offs between efficacy and efficiency.
>
> Edited:
> > While our discussion centres on physics-inductive biases, the same logic extends to other forms of prior knowledge: any regularising bias that constrains the learning task appropriately can potentially offer similar trade-offs between efficacy and efficiency. Identifying appropriate biases for other classes of problems, like in natural language processing or generative models, requires careful considerations of the specific tasks. The conclusions from our work should also hold in other domains if the right class of inductive biases are identified and used. Exploring these biases for other class of problems is outside the scope of this study.
>
>
> **Q1. No new conceptual understanding**
>
> Our contribution is not a new conceptual claim but a quantitative examination of the efficiency effects of physics biases under controlled, carbon-aware evaluation. There is little empirical work that measures these effects in terms of energy and carbon across diverse ML architectures.
>
>
> **Q2. Hyperparameter tuning**
>
> We refer to our response to W2. Both UNet and FNO were tuned to the same datasets including the one used in this work.
>
>
> **Q3. High inference cost of flow matching**
>
> Flow matching requires numerical integration at inference, which introduces multiple forward passes per prediction. This makes inference inherently more expensive than single-pass models such as U-Nets or FNOs. Our implementation follows standard flow matching practice, so the higher inference cost reflects the algorithmic structure rather than an implementation artefact. We will clarify this in the paper by adding the following in L408:
>
> > This follows from its reliance on numerical integration at inference, which requires multiple forward passes rather than a single pass.

---

### Official Review · Reviewer_4zZT · 2025-11-04

**Soundness:** 2
**Presentation:** 3
**Contribution:** 2
**Rating:** 2
**Confidence:** 3

**Summary:**

The paper claims that physics priors improve the carbon–accuracy balance for spatio-temporal forecasting. The experiments are a harmonic oscillator, viscous Burgers, and a small 2D periodic shear-flow subset. Carbon is computed with CarbonTracker using a fixed global grid intensity of 445 gCO2/kWh and a PUE factor; training is single-run; one-step inference is repeated three times. The main comparison is U-Nets, three FNOs, and a low-resolution flow-matching model that integrates 10 Euler steps per forecast.

My main concerns include the fact that the design fixes intensity and PUE, so reported CO2 is simply a constant multiple of energy/runtime. The "carbon" plots therefore do not add information beyond kWh and wall-time, so it is essentially a scalar multiple of what can be computed from existing descriptions of training runs and simulations. Moreover, it only computes this for their small-scale simulations, so likely does not accurately represent the tradeoff in real-world large-scale simulations. I think it would have been more accurately to simply perform a meta survey of existing simulations and models, rather than build them from scratch - because the observations are actually less realistic. In addition, the paper wholly misrepresents PINNs, and conflates them with amortised forecasters. In reality a PINN needs to be retrained for each new problem, and thus the author conclusions based upon them for forecasting tasks are invalid.

**Strengths:**

- It is helpful to report both training and inference footprints on the same page.
- The configs are stated clearly.

**Weaknesses:**

- The paper's framing of PINNs within a "forecasting" context is misguided. A PINN is essentially a solver that learns one specific PDE _solution_ via gradient descent, not an amortised forecaster. Retraining per instance means it cannot generalise to new initial conditions or rollouts. Plotting a PINN on the same accuracy–carbon Pareto axis as data-driven forecasters conflates fundamentally different paradigms. This therefore complicates their claim about physics-informed models, since the PINN was used as an example at one side of the spectrum, when in reality, it is a completely different approach. In addition, the PINN (marked as the most physics-informed model) is actually much more computationally expensive than data-driven forecasters, since it needs to be retrained for each new problem, which ends up providing evidence in the complete opposite direction of the main thesis of the paper. (Note that I don't doubt that more appropriate priors reduce computational expense - I am simply pointing out that a PINN is not a good example to use for making this claim)
- Because carbon intensity and PUE are fixed, CO2 is a scalar multiple of measured energy. The carbon plots therefore convey the same ranking as energy, though the authors do correctly measure real power usage (CarbonTracker) and note that runtime and energy are not strictly proportional. The carbon axis could be replaced by energy without loss of information. And the energy is closely tied to the GPU hours, which are already reported in many papers.
- The benchmarks are didactic and not representative. The Harmonic Oscillator and Burgers equations are teaching tools, not serious simulations. But you could have easily computed the carbon usage of large-scale simulations that had been run previously (since it is standard to report compute costs), which would give you a more realistic scenario. The shear-flow case is small, 2D, periodic. If the goal is to compare ML against simulation on carbon, use an established turbulence dataset (see, e.g., Johns Hopkins Turbulence Database) with known solver cost and compare solver vs surrogate at matched skill, rather than re-running toys and drawing conclusions based on a completely different size class.
- The FM setup is not matched to baselines. It operates at 1/16 resolution and performs 10 numerical steps per forecast, while others are full-resolution single-pass models. Any "bias vs carbon" statement gets confounded by resolution and integrator costs.
- The metrics do not test physics. Pearson r (plus VRMSE) says little about incompressibility, spectra, conservation, or error growth. This does not support a claim about “trading carbon for physics”.

**Questions:**

No questions. My concerns are fundamental rather than clarificatory. Please feel free to correct any mistakes in my understanding.

---

> ### Author Response · Authors · 2025-11-18
>
> We thank the reviewer for their time and effort in reviewing our paper. We would like to address the points they make in their review:
>
> **W1. PINNs**
>
> We thank the reviewer for the clarification. We agree that PINNs are optimisation-based PDE solvers rather than amortised forecasting models. In our work, PINNs appear only in the two small-scale PDE experiments (harmonic oscillator and viscous Burgers’ equation), where they are used to illustrate how physics priors affect efficacy and efficiency in controlled settings. They are not used as forecasters in the shear-flow study.
>
> We acknowledge that the distinction between solvers and forecasters was not made sufficiently clear in the current version of the paper. Their placement in the subsection “Spatio-Temporal Forecasting Models” and in Figure 3 is therefore misleading. We will revise both the subsection and the figure to make this separation explicit.
>
> Importantly, we do not place PINNs on the same accuracy–carbon Pareto axis as the data-driven forecasting models. Figure 1 compares a PINN and a NN only within the Burgers’ equation experiment, where the architectures are identical and differ solely in their loss formulation. This comparison is limited to the small-scale setting and is not intended as a forecasting comparison.
>
> **W2. Carbon vs. energy footprint**
>
> We fix carbon intensity and PUE to make model comparisons fair and independent of external factors such as time of day. Although carbon footprint is proportional to energy use under these assumptions, reporting carbon footprint remains important. It communicates the real environmental impact directly and supports carbon-aware evaluation, which energy alone does not fully capture. This point is also emphasised in e.g. [1].
>
> [1] Wright et al. Efficiency Is Not Enough: A Critical Perspective on Environmentally Sustainable AI, 2025. URL https://doi.org/10.1145/3724500
>
> **W3. Datasets**
>
> Our goal is not to compare ML surrogates against traditional solvers. The paper focuses on how physics-informed inductive biases shift the efficacy–efficiency trade-off relative to purely data-driven models. Established DNS datasets such as Johns Hopkins Turbulence Database are useful for benchmarking solver cost, but that is not the comparison we seek.
>
>
> **W4. FM model**
>
> We agree that the FM setup differs from the baselines because it follows a fundamentally different modelling paradigm. FM uses 1/16 resolution as a practical necessity, since a full-resolution implementation is computationally infeasible for this task. This is already noted on L438 and further detailed in Appendix E.
>
>
> **W5. Metrics**
>
> We agree that the metrics do not test physics and will note this in the limitations as follows:
>
> > The evaluation metrics focus on predictive accuracy rather than physical fidelity. Physics-based metrics would more directly assess whether the models capture underlying structures or conservation properties.

---

### Author Response · Authors · 2025-11-18
**Overall comment to all reviewers**

We thank all reviewers for their time and their thorough reviews. We are encouraged that they find the topic important and appreciate the depth and nuance of the analysis and discussion.

We emphasise that the study design reflects the primary message of the paper: promoting resource-awareness in ML experimentation. To remain consistent with this goal, we deliberately combine two small-scale studies with one large-scale experiment, demonstrating the accuracy–carbon trade-off without incurring unnecessary computational cost. While we agree that additional experiments could strengthen the message in the paper, we believe that the current evidence is adequate to foster discussions about resource-awareness and the use of physics inductive biases. We recognize this might not be what is expected based on the reviews, but we will stick to the key messaging in the paper and refrain from using additional compute to perform additional experiments.

There are several other clarifications that have been sought in the reviews, which we respond to in the individual responses. We look forward to engaging further in the author-reviewer discussions.

---

### Meta-Review · Area_Chair_Fbq6 · 2026-01-06

**Summary:**

The paper investigates whether physics-informed inductive biases improve the carbon–accuracy trade-off in spatio-temporal forecasting. It evaluates models ranging from purely data-driven to strongly physics-informed across three PDE-based benchmarks.

Fundamentally, there are flaws in the framing (i.e. PINNs were initially mischaracterised as forecasters and conflated with amortised models, obscuring the interpretation of the carbon–accuracy trade-off) and confunded comparisons (i.e. in resolution and inference procedure, complicating “physics vs carbon” findings).
The authors have addressed some of the concerns in the rebuttal, and claimed the paper will be further revised accordingly, but it is not clear if the paper has been revised as changes were not marked.

It seems that the reviewers were not excited about this paper and the fundamental flaws were not fully addressed.

**Reviewer Concerns:**

All the reviewers voiced strong concerns on this paper, both on the fundamental framing and contributions, methodology, and claims.
The authors addressed the clarity and framing issues, especially around PINNs, metrics, and hyperparameter tuning, and improve transparency in limitations. However, core structural limitations remain: reliance on synthetic benchmarks, lack of physics-based evaluation metrics, and conclusions that weaken as problem complexity increases.

**Reviewer Scores:**

I dont think there were any active discussions as the rebuttal was posted quite early. I dont believe the scores would have changed.

---

### Decision · Program_Chairs · 2026-01-26

Reject